# The Effect of Supplementary LED Lighting on the Morphological and Physiological Traits of Miniature *Rosa* × *Hybrida* ‘Aga’ and the Development of Powdery Mildew (*Podosphaera pannosa*) under Greenhouse Conditions

**DOI:** 10.3390/plants10020417

**Published:** 2021-02-23

**Authors:** Bożena Matysiak

**Affiliations:** The National Institute of Horticultural Research, Department of Applied Biology, 96-100 Skierniewice, Poland; bozena.matysiak@inhort.pl

**Keywords:** chlorophyll *a* fluorescence, flavonols, spectral light quality, nitrogen balance index, morphological traits

## Abstract

We investigated the growth traits, flower bud formation, photosynthetic performance, and powdery mildew development in miniature *Rosa* × *hybrida* ‘Aga’ plants grown in the greenhouse under different light-emitting diode (LED) light spectra. Fluorescence-based sensors that detect the maximum photochemical efficiency of photosystem II (PS II) as well as chlorophyll and flavonol indices were used in this study. Five different LED light treatments as a supplement to natural sunlight with red (R), blue (B), white (W), RBW+FR (far-red) (high R:FR), and RBW+FR (low R:FR) were used. Control plants were illuminated only by natural sunlight. Plants were grown under different spectra of LED lighting and the same photosynthetic photon flux density (PPFD) (200 µmol m^−2^ s^−1^) at a photoperiod of 18 h. Plants grown under both RBW+FR lights were the highest, and had the greatest total shoot length, irrespective of R:FR. These plants also showed the highest maximum quantum yield of PS II (average 0.805) among the light treatments. Red monochromatic light and RBW+FR at high R:FR stimulated flower bud formation. Moreover, plants grown under red LEDs were more resistant to *Podosphaera pannosa* than those grown under other light treatments. The increased flavonol index in plants exposed to monochromatic blue light, compared to the W and control plants, did not inhibit powdery mildew development.

## 1. Introduction

Roses (*Rosa* sp.) are the most important ornamental plants worldwide. The EU is the most important production area for ornamental plants in the world, and holds the first place in cut flower and potted plants with 31.0% of the global value [1]. EU production of flowers and ornamental plants in 2019 was worth 22.099 million Euros. Roses are the number one cut flower on the European market, and potted roses are among the top five best-selling potted plants. Light is a key factor affecting the growth and flowering of roses. In northern Europe, the relatively low natural light intensity and short days during winter result in fewer axillary shoots, lower fresh weight and diameter of flower stems and buds, lower leaf area, lower number of petals per flower, and poorer pigmentation [2,3]. Furthermore, flower buds may be aborted under low radiation, resulting in blind shoots and a low post-harvest quality. In temperate climate zones, greenhouse-grown roses therefore require supplementary lighting during winter.

Light-emitting diode (LED) lamps have attracted increased interest as a primary source of irradiation, or as supplementary lighting, for plants due to their high photosynthetically active radiation (PAR) efficiency, low energy consumption, long life, and narrow emission spectrum [4,5,6,7,8]. Another unique aspect of LED light is the possibility of setting a desirable spectral composition for the targeted manipulation of metabolic responses to optimise plant photosynthesis and morphogenesis, as well as primary and secondary metabolite metabolism [9,10,11]. Manipulation of the light environment may provide an alternative strategy for suppressing pathogens while maintaining plant health and productivity [12].

Light quality influences all aspects of plant biology, and much is known about how different plant species respond to the light environment [13]. However, the mechanisms of adjusting the photosynthetic activity of roses to the quality of light, and the modulation of plant architecture through qualitative light treatments are not fully understood. Red light (640–680 nm) strongly stimulates leaf photosynthesis and light absorptance in the leaves and canopy of *Rosa* plants [14], whereas blue light (400–525 nm) is more involved in chlorophyll synthesis, and increases stomatal conductance and intercellular CO_2_ concentration [15]. A certain quantity of blue light is required for the functional photosynthetic and physiological reaction of roses [9,16,17]. Monochromatic red and/or blue LEDs are generally used to evaluate the impact of light quality on biomass production, photosynthetic performance, and morphology of roses. However, the results of studies conducted on other horticultural plant species indicate that plants grown under supplemental multi-wavelength irradiation or white light show higher photosynthetic activity than those grown under monochromatic light [18,19]. Growing *Rosa* plants under monochromatic red and blue lights can reduce the plants’ ability to cope with high-light-intensity stress compared to red and blue lights enriched with white LEDs [20]. Furthermore, light quality affects morphogenesis and overall plant appearance, but the phenotypic responses to red, far-red, and the ratio of red to far-red (R:FR), can vary among species and growing conditions [21,22].

Powdery mildew, caused by the obligate biotrophic fungal pathogen *Podosphera pannosa*, is one of the most frequent and serious diseases of greenhouse-grown roses [23]. The favourable temperature and relative humidity in greenhouses are the primary drivers of powdery mildew epidemics. Chemical fungicides have mainly been used for years to manage this disease. However, increasing public concern about the use of fungicides calls for the development of non-chemical control methods. Recent studies indicated that the modification of the light spectrum emitted by LEDs has great potential in disease control in greenhouses [24,25]. Previous studies have shown that exposing plants to red light stimulates resistance to a number of pathogens compared to plants exposed to sunlight and other spectra [26,27,28]. Other authors indicated a significant influence of the R:FR and blue light on disease development [24,29]. A limited number of studies have focused on finding a practical use of light quality to control powdery mildew on *Rosa*, focusing on the effect of visible light and UV-B radiation on *Podosphera pannosa* under laboratory conditions [30], as well as monochromatic blue, red, and far-red LEDs under greenhouse conditions, but at a low photosynthetic photon flux density (PPFD; 5 µmol m^−2^ s^−1^), and with a positive effect of the red spectrum [31].

Flavonols, a class of flavonoids, are thought to play important roles in plant defence against abiotic stresses (UV radiation) and phytopathogens because of their antimicrobial activity and local accumulation after fungal infection [32,33]. Light is one of the most important environmental factors affecting flavonoid biosynthesis in plants [34,35], and the respective pathway is strongly regulated by UV, while the spectra in the red and blue regions also have a significant influence. However, the impact of light quality on the production of flavonoids in plants is species- and compound-specific. The accumulation of flavonols in plants can be assessed in a non-destructive manner, which is particularly useful for the quick analysis of greenhouse-grown horticultural plants [36].

The objectives of this study were to investigate the growth traits, flower bud formation, photosynthetic performance, and powdery mildew occurrence in *Rosa* ‘Aga’ plants grown in a greenhouse under different light spectra emitted by LEDs. *Rosa* ‘Aga’ was selected due to being highly susceptible to powdery mildew [37]. Fluorescence-based sensors that detect chlorophyll and flavonol indices, as well as the maximum photochemical efficiency of PS II were used in this study. The determination of chlorophyll *a* fluorescence is a non-invasive measurement of PS II activity and photosynthetic performance, and commonly used in plant physiology. Ahlman et al. [38] indicated the suitability of this method for optimizing the LED light spectrum for plants. The overall aim of the work was to improve crop management strategies for miniature rose cultivation in greenhouses in northern climates, where supplemental LED lighting is used.

## 2. Results and Discussion

### 2.1. Plant Growth and Morphology 

Supplemental lighting with different light spectra generated by LEDs and equal PPFD (200 μmol m^−2^ s^−1^) significantly affected the growth rate and flower bud formation of the miniature rose ‘Aga’ grown in a greenhouse under insufficient sunlight. This influence was significant after just 1 week of LED lighting. After 6 weeks, the plants grown under a supplemental wide light spectrum, generated jointly by red, blue, white, and far-red LEDs (S1 and S2 treatments), were 27% higher than the control plants illuminated with daylight only, 18% higher than the plants grown under supplemental white LEDs, and 9% higher than those under supplemental R and B LEDs (Figure 1).

The total shoot length of plants exposed to S1 and S2 spectra was 53% higher than that of control plants, and on average 21% higher than that of plants under R and B LEDs (Figure 2). Monochromatic red and blue LEDs, as well as white LEDs, were not as effective in controlling plant growth as S1 and S2 treatments. However, plant height and total shoot length of plants exposed to R or B LEDs were on average 20% and 43%, respectively, higher than those of the control plants. No significant differences were found in plant height and total shoot length in plants under monochromatic R and B LEDs.

Red monochromatic LEDs significantly stimulated the development of adventitious buds in the studied miniature roses, ‘Aga’. Plants grown under supplemental red LEDs had the largest number of shoots per plant (Figure 3).

The formation of flower buds was strongly dependent on the quantity and quality of LED lighting. After 6 weeks, plants grown under daylight formed only 1.5 flower buds and flowers per plant, while those irradiated with supplemental LEDs formed 4 to 5.5, depending on the light spectrum. In the early growth stages, the S1 and S2 spectra had a significant, positive effect on the formation of flower buds, and a slightly lower one with supplemental R light only (Figure 4). However, after 6 weeks, both monochromatic red light as well as the S1 spectrum with a high R:FR ratio most significantly stimulated the formation of flower buds.

Our results are generally consistent with those of other studies on the responses of *Rosa* plants to R and FR. For example, an decreased R:FR reduced the percentage of sprouted buds in *Rosa* [21]. On the other hand, in our study R:FR in a wide range of ratios from 2.1 to 11.3 had no effect on the elongation of *Rosa* ‘Aga’ shoots. An increase in stem length, in the number of axes, and in the number of flowered axes of *Rosa* was observed under the far-red enriched light, and a strong genotype–light-quality interaction, i.e., a genotype-specific response was observed by Crespel et al. [39].

### 2.2. Chlorophyll Fluorescence

We showed that the miniature *Rosa* ‘Aga’ requires a different spectrum of light at different developmental stages to obtain the maximum quantum yield of photosystem II (PSII), as measured by variable to maximum chlorophyll (Chl) fluorescence (F_V_/F_M_)_._ After 1 week, the highest F_V_/F_M_ values were found in plants growing under B, as well as S1 and S2 spectra (average 0.804), after 3 weeks under B (0.804), while the F_V_/F_M_ values for the control plants were the lowest (0.790) for both dates (Figure 5). After 6 weeks, the plants growing under S1 and S2 light treatments showed the highest maximum quantum yields of PS II (average 0.805) among the light treatments. The lowest F_V_/F_M_ values, in the range of 0.773–0.785, were observed in control plants, as well as under R, B, and W LEDs. An F_V_/F_M_ value in the range of 0.79 to 0.84 is the approximate optimal value for many plant species, with lowered values indicating plant stress [40]. There is limited information on the effect of light quality on photosynthetic capacity and chlorophyll fluorescence emissions for *Rosa* plants. Paradiso et al. [14] showed the spectral dependence of leaf photosynthesis and light absorptance in leaves and canopy of rose in the short term. The highest values of the action spectrum were observed under red light (peaking at 660–680 nm), with a relatively high maximum in the blue region (445 nm), and a broad minimum was found in the green region (500–580 nm). The photosynthesis rate declined rapidly above 680 nm, with an extremely low photosynthesis at 720 nm. Here, we found that light qualities significantly influenced the maximum quantum yield of photosystem II (PS II). Multi-wavelength irradiation (S1 and S2 treatments) with a relatively high share of red light (49–67%), in combination with blue (13–16%) and green-yellow fractions of visible light (10–15%), was most conducive to efficient photosynthesis and performance in *Rosa* leaves, regardless of the R:FR ratio (in the range from 2.1 to 11.3). Moreover, these plants were characterised by the strongest growth, with the highest stem length and total shoot length.

Similar results, showing the positive influence of the combined use of red and blue light on maximum quantum yield of PSII, compared to solely blue or red, have been described for some horticultural plants such as *Dendrobium officinale* [41], *Lactuca sativa* [42], *Solanum lycopersicum* [19], and *Campomanesia pubescens* [43], as well as *Cordyline australis*, *Ficus benjamina*, and *Sinningia speciosa* [44]. At the same time, other authors indicated that monochromatic red light reduced the photosynthetic efficiency, and led to photo-damage after long-term exposure [9,43,44]. In another study, F_V_/F_M_ for *Houttuynia cordata* seedlings was the highest under blue LED light, and the lowest under green LED light [45]. For *Phalaenopsis*, F_V_/F_M_ was overall slightly higher under white LEDs, and significantly lower under monochromatic red and combined red and blue LEDs [46]. These results indicate that the different wavelengths of LED lighting specifically regulate the efficiency of PSII photochemistry, and this reaction is genotype-dependent.

### 2.3. Chlorophyll, Flavonols, and Nitrogen Balance Indices

The chlorophyll content index (CCI) in plants after 6 weeks of growth was not dependent on the spectrum of supplementary light emitted by the LEDs (Figure 6). Previous studies have reported the differential response of light quality on leaf chlorophyll content. Long term exposure of leaves to solely blue or a high share of blue in the light spectrum was favourable for chlorophyll content in *Cordyline australis*, *Sinningia speciose*, and *Lactuca sativa*, and no effects on chlorophyll content were found for *Ficus benjamina* [42,44]. However, for *Rosa* ‘Avalanche’ chlorophyll *a* and *b*, and total chlorophyll content significantly decreased by growing rose plants under monochromatic blue light compared to red, red and blue (70:30%), and white light [20]. Similar results was also found for *Rosa* ‘Radrazz’ and *Nicotiana tabacum*, where the chlorophyll content of plants decreased under blue light, and increased under red light or white light [15,19]. The variability in light quality responses among crop genotypes could be caused by leaf morphology and mesophyll anatomy [44], as well as the experimental environments used, including light intensity and photoperiod. Although the index of chlorophyll content index in *Rosa* ‘Aga’ was not dependent on the light quality in our study, the maximum quantum yield of PSII was the highest under multi-wavelength irradiation (S1 and S2 treatments) and the lowest under red, blue, and white LED light after 6 weeks of irradiation, while the plants reached commercial size. Our study revealed that plants in mature stage showed a more sensitive electron transport system under monochromatic red and blue or white LED as compared to those under multi-wavelength irradiation. Moreover, the maximum quantum efficiency of PS II photochemistry (F_V_/F_M_ parameter) was a more sensitive parameter, compared to chlorophyll content index, to explore light stress in roses.

Plants exposed to monochromatic blue light showed the highest flavonol index (0.8), which was higher than for plants grown under natural light, and exposed to white LEDs, by 17% and 8%, respectively. There are many scientific reports showing the stimulatory effect of blue light on the synthesis of epidermal flavonoids in horticultural plants [10,35,47,48,49]. For example, greenhouse-grown miniature *Rosa* ‘Scarlet’ under LED arrays had the highest flavonoid content at the ratio of blue to red of 40 to 60%, followed by 20 to 80%; the lowest value was found for white LEDs [9]. The results obtained by the non-invasive method with Dualex in this study were positively correlated with those obtained via high pressure liquid chromatography–mass spectrometry (HPLC-LCMS). As blue light increased, the rutin and quercetin contents in the *Rosa* leaves increased [9].

The highest nitrogen balance index (48) and the lowest flavonol index (0.67) in our study were recorded for the control plants. Significantly lower nitrogen balance indexes and higher flavonol indexes were recorded for irradiated plants than for the control plants, however the influence of different LED light spectra on the nitrogen balance index (NBI) was not demonstrated. Flavonoids, as non-nitrogenous secondary metabolites, are considered indicators of nitrogen availability in a plant [50,51]. The flavonoid content increases under low N availability [36,51,52]. Roses in our study were equally fertigated, therefore, it can be assumed that the light conditions influenced the N metabolism. It is known that nitrate reductase enzyme, which catalyzes the reduction of nitrate to nitrite, is activated by light, and inactivated in darkness conditions [53]. The enzyme activation after exposition to high photosynthetically active radiation (PAR) led to a decrease of nitrate concentration in *Brassica* leaves [54]; conversely, its inactivation caused nitrate accumulation in plant tissues. In our case, it is possible that under a poorly lit environment (control treatment), the nitrate reductase activity decreased, allowing for the accumulation of nitrate that could be used in flavonoid synthesis. On the contrary, the enzyme activation after exposing plants to LED lights led to a decline of nitrate concentrations in leaves, and to a reduction of the nitrogen balance index. Our study showed no significant difference between the nitrogen balance index for plants exposed to different light spectra at PPFD 200 μmol m^−2^ s^−1^. However, at a lower PPFD, the light spectrum can modify the enzyme activity and consequently affect the nitrogen balance index, as previously demonstrated by Silvestri [55].

### 2.4. Disease Evaluation

The first symptoms of powdery whitish growth of the mildew fungus, caused by *Podosphaera pannosa*, on *Rosa* ‘Aga’ leaves occurred at 1 week of growth in the greenhouse (Figure 7).

Red monochromatic LEDs (660 nm) at 200 µmol m^−2^ s^−1^ PPFD reduced the severity of powdery mildew on *Rosa* ‘Aga’. After 6 weeks, the symptoms of powdery mildew were less severe under red LEDs (average score 1.6, about 3% of leaf area covered with mycelium) than under the control, white LEDs, S1, and S2 treatments (average score 3.0, up to 10% of leaf area covered with mycelium). Red light enriched with blue, green, and far-red spectra, i.e., S1 and S2 treatments, as well as the wide spectrum of white LEDs did not affect plant disease development as strongly as monochromatic red light. The inhibitory effect of red light on powdery mildew development was observed just 1 week after the plants were exposed to LED lighting. Our results suggest a role of red light in the light-enriched resistance of *Rosa* ‘Aga’ to *P. pannosa*. Previous studies conducted by other authors demonstrated that interrupting the 6-h dark period with 1-h exposure to red light at low intensity (5 µmol m^−2^ s^−1^) was as effective as continuous light in supressing the productivity of conidia; however, the positive effect of red light was reversed by 1-h of far-red light [31]. Our results show that red monochromatic LED light used at high PPFD (200 μmol m^−2^ s^−1^) for 18 h effectively inhibits the development of powdery mildew on *Rosa* ‘Aga’ in greenhouse conditions. However, R:FR in the range of ratios 2.1–11.3 did not reverse this effect. Reduced pathogen resistance by low R:FR has been observed in some horticultural plants, and was correlated with the downregulation of the jasmonate and salicylic acid signalling pathways [21,56].

Suthaparan et al. [31] found that blue LED light at low intensity decreased *P. pannosa* conidia germination on detached leaflets of *Rosa* compared to white light, but it did not reduce conidium development with 18 h of daylight supplemented with 6 h of blue light. Our study shows that plants grown under monochromatic blue LED light at 200 μmol m^−2^ s^−1^ and an 18-h photoperiod were most affected by powdery mildew (score 4.2), and the symptoms were more severe than under white LEDs and the control treatment. Moreover, these plants accumulated the greatest amount of flavonols in leaves, as the FLAV index was highest. These data indicate that the amount of accumulated flavonols in leaves was insufficient to induce the inhibitory effect of the pathogen by blue light, despite the antimicrobial activity of flavonoid compounds. Recent scientific reports indicate that inoculation of detached leaves of the susceptible *Rosa* cultivar with *P. pannosa* led, not only to the general pathogen response, but also to a downregulation of genes related to photosynthesis [57]. We could not show a close relationship between photosynthetic performance (F_V_/F_M_) and the nitrogen status of plants (NBI) and disease severity of powdery mildew on *Rosa* ‘Aga’ after 6 weeks of plant growth.

## 3. Materials and Methods

### 3.1. Plant Material and Growth Conditions

The potted miniature rose (*Rosa* × *hybrida*) ‘Aga’, which is highly susceptible to powdery mildew [45], was obtained from a commercial nursery (HRS Dawidy, Warsaw, Poland). The experiment was carried out in a high-tech greenhouse of The National Institute of Horticultural Research in Skierniewice, Poland (51°57′ N, 20°08′ E). Plants were grown in peat substrate (Klasmann-Deilmann, Geeste, Germany) TS1, pH 6.0, N 140, P 70, K 150, Mg 38 (in mg/L and microelements) with four rooted cuttings, pinched once (in the nursery), per one 12-cm pot. The cuttings were 10–12 cm long and had two emerging leaves. Pots were placed on ebb-and-flow benches (1.2 × 2 m) at a spacing of 20 plants per m^2^. The benches with different light treatments were separated with white lightproof screens. Fertigation was carried out twice a week with a water solution of Kristalon Red 12 + 12 + 36 Yara (N 12%, P 12%, K 36%, S 1%, Fe 0.07%, Mn 0.04%, Zn 0.025%, Cu 0.025%, Mo 0.004%) fertiliser at a concentration of 0.1% (EC 1.2 ms cm^−1^). The temperature during the day/night was set to 21/18 °C and the relative humidity to 60–65%. The experiment was started on 11 March 2019, and ended after 6 weeks, when the plants achieved commercial value.

### 3.2. Light Treatments

The greenhouse chamber (5 × 4 m) was equipped with purpose-built LED arrays containing diodes (Osram, Munich, Germany). Five LED light treatments with: (1) R—red (660 nm); (2) B—blue (450 nm); (3) W—white (cool-white diodes, 5000 Kelvin); and two different colour-mixtures of R, B, W and FR—far-red (730 nm), such as; (4) S1—RBW + FR (high ratio R:FR); and (5) S2—RBW + FR (low ratio R:FR), were used. The relative spectra of the light treatments are shown in Table 1 and Figure 8. Plants were grown under supplemental LED lighting of different spectra at the same overall PPFD of 200 µmol m^−2^ s^−1^ at plant level for 18 h per day, so that the daily light integral (DLI) inside the greenhouse was increased by 12.96 mol m^−2^ d^−1^. The control plants were grown in daylight (natural day length 11 h) only, without any supplemental lighting. The average DLI inside the greenhouse (natural sunlight) during the experimental period was 7.5 mol m^−2^ d^−1^. The PPFD measurements and spectral quantification were performed using a spectrometer GL Spectrolux (GL Optic, Puszczykowo, Poland, https://gloptic.com accessed on 12 December 2020).

### 3.3. Growth Parameters

The measurements were conducted three times during the 6-week experimental period. For this, 30 plants were randomly selected for growth parameters within each treatment. Plant height, number of shoots per plant, total shoot length per plant, and number of flower buds and flowers were evaluated after 1, 3, and 6 weeks.

### 3.4. Chlorophyll, Flavonol, and Nitrogen Balance Indices

We used an optical sensor for the assessment of chlorophyll and flavonol compounds, based on the measurement of UV absorbance of the leaf epidermis by the double excitation of chlorophyll fluorescence (Dualex Scientific + instrument, Force-A, Orsay, France, https://www.force-a.com accessed on 12 December 2020). The nitrogen balance index (NBI) was automatically calculated as the ratio of the chlorophyll index (CCI) to the flavonol index (FLAV), i.e., NBI = CCI/FLAV. The device used in this study allows for non-destructive measurements of chlorophyll, flavonol contents, and nitrogen balance in leaves, which makes it particularly suitable for photophysiological research [36]. For each light treatment, 30 young, fully expanded leaves were used for the assessment of the flavonol and chlorophyll indices.

### 3.5. Chlorophyll Fluorescence Parameters

Chlorophyll fluorescence, as an indicator of photosynthetic reactions, was analysed using a Mini PAM (Walz, Effeltrich, Germany, https://www.walz.com accessed on 12 December 2020). For each light treatment, chlorophyll a fluorescence emission from the upper leaf surface of 30 intact, dark-adapted leaves (30 min) was measured. After dark adaptation, the fluorescence variables (F_V_), minimal fluorescence (F_0_), maximum fluorescence (F_M_), and F_V_/F_M_, were determined. The minimal (F_0_) fluorescence was recorded with modulated low-intensity light below 0.1 µmol m^−2^ s^−1^, without affecting the variable fluorescence. Maximum (F_M_) fluorescence was estimated by an 0.8-s long saturation light pulse (2600 µmol m^−2^ s^−1^) with a 20,000-Hz frequency. The variable fluorescence (F_V_) was calculated by the equation F_V_ = F_M_ − F_0_. The F_V_/F_M_ ratio was obtained from the F_V_ and F_M_, and represents the potential maximal photochemical efficiency of PS II.

### 3.6. Disease Evaluation

The powdery mildew pathogen survives in greenhouses around the year, and therefore, infection occurs naturally in susceptible cultivars such as *Rosa* ‘Aga’. Plants at stage two, with emerging leaves, were transferred to the greenhouse chamber with high *Podosphera pannosa* inoculum potential. The experiment was carried out in a greenhouse chamber where ‘Aga’ roses had been grown previously. Horizontal airflow fans were used to ensure air mixing and a stable spore distribution over the course of the experiment. The data on disease severity were recorded three times (after 1, 3, and 6 weeks) under natural infection by following a disease rating based on a 0–5-scale development for the estimation of rose powdery mildew, according to Wojdyła [58]; 0—no symptoms, 1—up to 1% of shoot/leaf surface area covered with mycelium, 2—from 1.1 to 5%, 3—from 5.1 to 10%, 4—from 10.1 to 20%, 5—over 20% of shoot/leaf surface area covered with mycelium and abundant conidia. We evaluated 30 plants in each light treatment for each term.

### 3.7. Statistical Analysis

The experiment comprised a total 180 pots (30 for each of the 6 light treatments) with 4 plants per pot (total 720 plants). Statistical analysis was performed using the STATISTICA software, version 13.1 (StatSoft Inc., Tulsa, OK, USA). Data were analysed using one-way analysis of variance (ANOVA), and the treatment means were compared using Tukey’s HSD test at α = 0.05.

## 4. Conclusions

The modern technology of LED lighting allows producing high-quality potted roses grown in greenhouses under unfavourable light conditions, and suppressing powdery mildew development (*Podosphaera pannosa*). An optimal effect related to maximizing plant quality (plant size, number of flowers, brunching) and minimizing powdery mildew infestation can be obtained with a relatively high share of red light (49–67%) in combination with blue (13–16%) and green–yellow fractions of visible light (10–15%) at a PPFD of 200 μmol m^−2^ s^−1^ for 18 h, and at an R:FR ratio from 2.1 to 11.3. The positive effect of this LED light recipe on plant productivity and health may result from effective photosynthesis and synthesis of secondary metabolites. Modification of light quality might be an alternative to chemical fungicides to control powdery mildew in *Rosa* plants.

## Figures and Tables

**Figure 1 plants-10-00417-f001:**
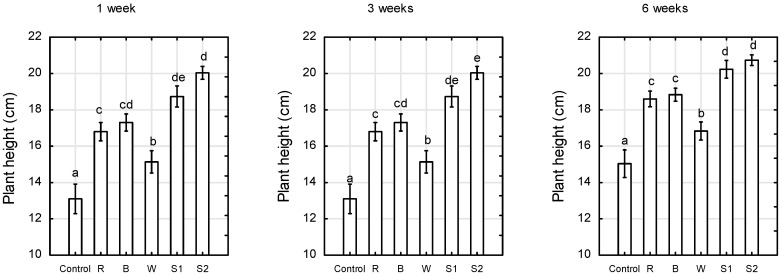
Plant height of *Rosa* ‘Aga’ after 1, 3, and 6 weeks of growth under six different light-emitting diode (LED) light treatments (Control—unlighted, R—red, B—blue, W—white, S1—RBW + FR (far-red) with high ratio of red to far-red (R:FR), and S2—RBW + FR with low R: FR. Average values (*n* = 30, ±SE) followed by different letters differ significantly according to Tukey’s test (*p* < 0.05).

**Figure 2 plants-10-00417-f002:**
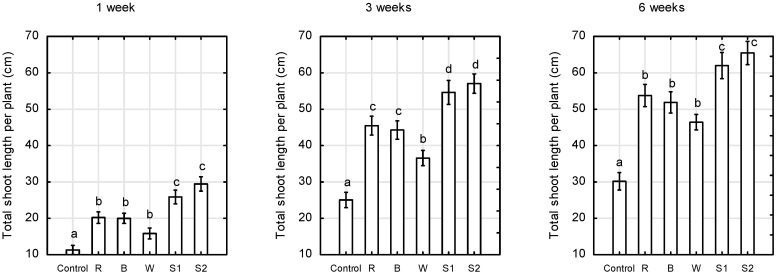
Total shoot length per plant of *Rosa* ‘Aga’ after 1, 3, and 6 weeks of growth under six different light-emitting diode LED light treatments (Control—unlighted, R—red, B—blue, W—white, S1—RBW + FR with high R:FR, and S2—RBW + FR with low R:FR. Average values (*n* = 30, ±SE) followed by different letters differ significantly according to Tukey’s test (*p* < 0.05).

**Figure 3 plants-10-00417-f003:**
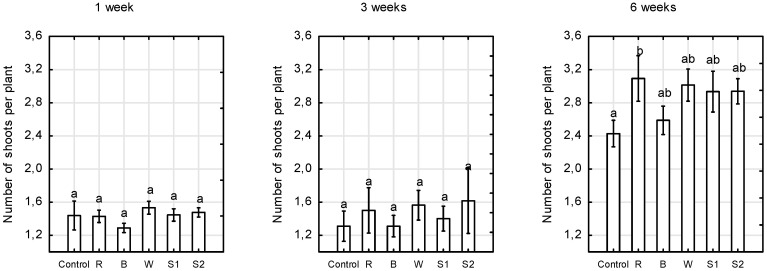
Number of shoots per plant of *Rosa* ‘Aga’ after 1, 3, and 6 weeks of growth under six different light-emitting diode LED light treatments (Control—unlighted, R—red, B—blue, W—white, S1—RBW + FR with high R:FR, and S2—RBW + FR with low R:FR. Average values (*n* = 30, ±SE) followed by different letters differ significantly according to Tukey’s test (*p* < 0.05).

**Figure 4 plants-10-00417-f004:**
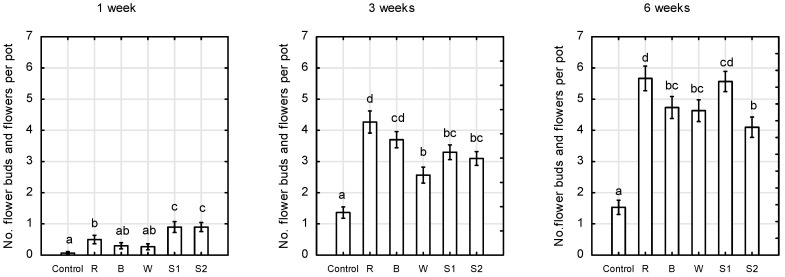
Number of flower buds and flowers per pot of *Rosa* ‘Aga’ after 1, 3, and 6 weeks of growth under six different light-emitting diode LED light treatments (Control—unlighted, R—red, B—blue, W—white, S1—RBW + FR with high R:FR, and S2—RBW + FR with low R:FR. Average values (*n* = 30, ±SE) followed by different letters differ significantly according to Tukey’s test (*p* < 0.05).

**Figure 5 plants-10-00417-f005:**
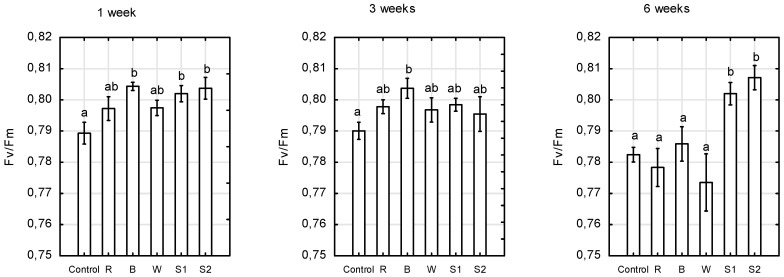
Changes in chlorophyll *a* fluorescence parameters—the maximum quantum yield of PSII, represented by F_V_/F_M_ (the variable fluorescence F_V_ to maximum fluorescence F_M_) in *Rosa* ‘Aga’ after 1, 3, and 6 weeks under six different light-emitting diode LED light treatments (Control—unlighted, R—red, B—blue, W—white, S1—RBW + FR with high R:FR, and S2—RBW + FR with low R:FR. Average values (*n* = 30, ±SE) followed by different letters differ significantly according to Tukey’s test (*p* < 0.05).

**Figure 6 plants-10-00417-f006:**
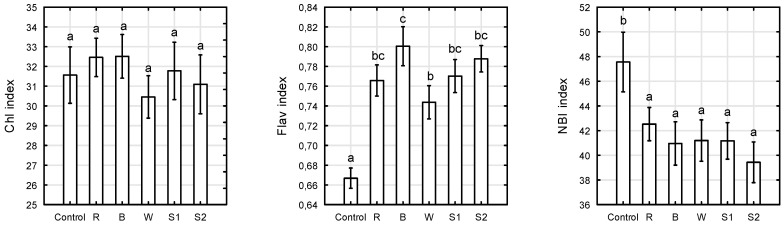
Chlorophyll content, flavonols, nitrogen balance index (NBI) indices in *Rosa* ‘Aga’ after 6 weeks of growth under six different light-emitting diode LED light treatments (Control—unlighted, R—red, B—blue, W—white, S1—RBW + FR with high R:FR, and S2—RBW + FR with low R:FR. Average values (*n* = 30, ±SE) followed by different letters differ significantly according to Tukey’s test (*p* < 0.05).

**Figure 7 plants-10-00417-f007:**
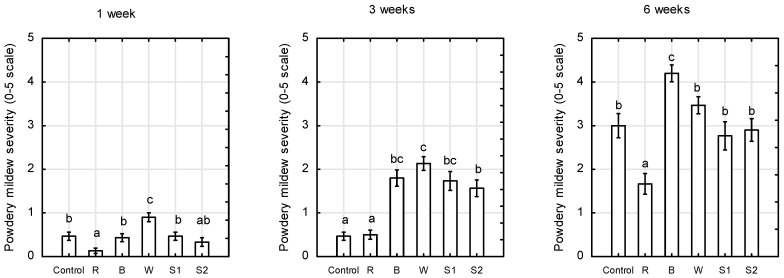
Powdery mildew severity (0–5-scale) in plants of *Rosa* ‘Aga’ after 1, 3, and 6 weeks of growth with natural infection in greenhouses, and grown under six different light-emitting diode LED light treatments (Control—unlighted, R—red, B—blue, W—white, S1—RBW + FR with high R: FR and S2—RBW + FR with low R:FR. Average values (*n* = 30, ±SE) followed by different letters differ significantly according to Tukey’s test (*p* < 0.05).

**Figure 8 plants-10-00417-f008:**
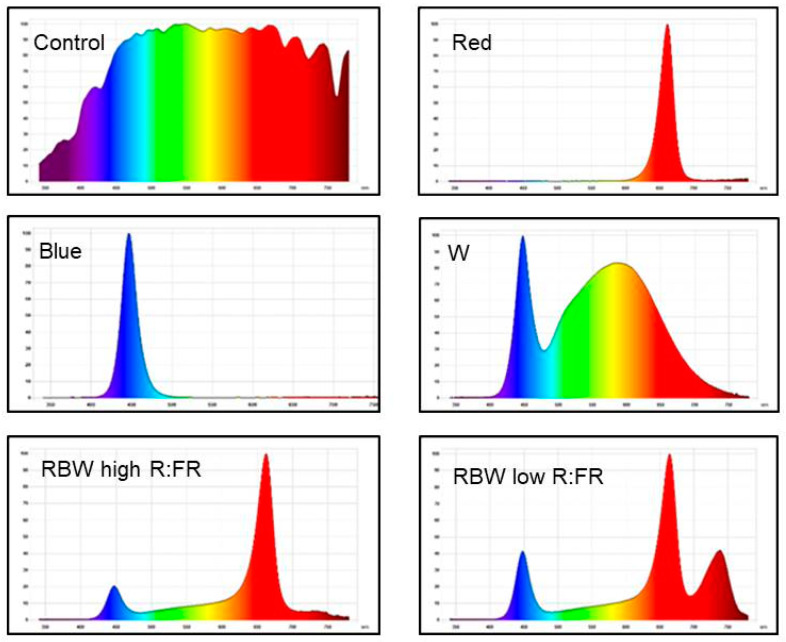
Spectral photon flux distribution (Y axis: relative values from 0 to 100%) for 340- to 780-nm lighting treatments. Charts in the upper row from left to right: control—sunlight inside greenhouse, R—red. Charts in the middle row from left to right: B—blue, W—white. Charts in the lower row from left to right: S1—RBW + FR with high R:FR and S2—RBW + FR with low R:FR.

**Table 1 plants-10-00417-t001:** Spectral distribution (%) for the five light-emitting diode LED light treatments and control, natural sunlight in greenhouse (fraction of integral photon flux ranging from 340 to 780 nm in ultraviolet, blue, green-yellow, red, and far-red). Spectra were recorded and averaged at six locations.

LED Light Treatments	UV-A 340–400 nm	Blue 400–500 nm	Green-Yellow 500–600 nm	Red 600–700 nm	Far-Red 700–780 nm	R:FR Ratio
Control	3.9	22.1	28.0	27.4	16.8	1.6
R—Red	-	-	-	100.0	-	-
B—Blue	-	100.0	-	-	-	-
W—White	0.1	24.4	44.4	28.9	2.2	13.1
S1—R + B + W + FR high	0.3	12.9	14.5	66.5	5.9	11.3
S2—R + B + W + FR low	0.2	16.3	10.3	49.4	23.8	2.1

## Data Availability

The data presented in this study are available on request from the corresponding author.

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
