# Peer review of "The Effect of Supplementary LED Lighting on the Morphological and Physiological Traits of Miniature Rosa × Hybrida ‘Aga’ and the Development of Powdery Mildew (Podosphaera pannosa) under Greenhouse Conditions"

_plants, 2021, doi:10.3390/plants10020417_

Round 1
Reviewer 1 Report
The manuscript “Modulation of the LED light spectrum to improve the growth and quality of miniature Rosa × hybrida ‘Aga’ and inhibit the development of powdery mildew (Podosphaera pannosa) under greenhouse conditions” provides insight knowledge about Effects of different LED light spectra on the growth and quality of Rose. However, the main objective of this study is still unclear which led this experiment and its future prospects as well because the author used only one variety susceptible to powdery mildew, instead of one resistant variety as well which could pace better results to compare among resistant and susceptible varieties for better understanding the efficiency of light spectra.
Overall, this manuscript is well written and thoroughly described. However, the language used in the manuscript is not up to the standard for publication, and several major and minor mistakes are shown, and many of the phrases are not clear. It should go through a complete proof-reading by a native speaker.
Results and Discussion section needs serious revisions because it needs more precise and strong justifications for the results obtained during the experiments. Moreover, there are many sentences and grammatical errors throughout this section. Most of the results are very speculative and then following discussions are mostly irrelevant and need solid reasons for the results obtained. There are some irrelevant and outdated literature references that does not suit to the arguments provided by the author.
Please see this article to better understand the discussions section “Chlorophyll, flavanols and nitrogen balance indices”. say something about this article to relate your results.
Silvestri C, Caceres ME, Ceccarelli M, Pica AL, Rugini E and Cristofori V. (2019). Influence of Continuous Spectrum Light on Morphological Traits and Leaf Anatomy of Hazelnut Plantlets. Front. Plant Sci. 10:1318. doi: 10.3389/fpls.2019.01318.
Some other minor comments are as follows
The title is confusing and should be revised in a proper scientific manner.
Line 9: We investigated the growth traits,
Line 11: greenhouse under different LED light spectra.
Line 13: Five different LED light treatments as supplement to natural sunlight
Line 14: Control plants were not illuminated…. I think this is not true. If you enrich the natural light with other different spectra, I think the control is illuminated by natural sunlight. Or not?
Keywords: Provide some more precise keywords
Provide the latest statistical worldwide data for area and production of Rose in introduction
The introduction section needs some serious revisions as there are some irrelevant literatures. Develop a precise relationship with your hypothesis in introduction section along with some latest relevant literature. Also provide specific future prospects of this study at the end of introduction.
Line 279: with four rooted cuttings, in 12-cm pot each and once pinched. (pinching was performed at how many days after transplanting?)
In summary, presented data show a positive scenario regarding growth and quality of rose under different light spectra, but discussion of results needs serious revisions. To sum up, the manuscript can find interest among specialists when these few comments will be taken into account.
Author Response
First of all, I would like to thank the referee whose suggestions have definitely improved the paper.
The title of the article was changed to be more precise and referring to the aim of the study and results. Currently, the tittle is “The effect of supplementary LED lighting on the morphological and physiological traits of miniature Rosa × hybrida ‘Aga’ and the development of powdery mildew (Podosphaera pannosa) under greenhouse conditions”.
In our study we used only one variety which is sensitive to powdery mildew and therefore the name of the variety ‘Aga’ appears in the title of the article. We assessed the development of powdery mildew in plants under different LED light spectra. If we had used a resistant variety, we would not have had such a possibility, although it would be a really important scientific aspect as suggested by the reviewer. The health aspect of roses is very important, and powdery mildew is one of the most dangerous and common diseases in greenhouse cultivation. All environmentally friendly and innovative solutions aimed at limiting the occurrence and development of powdery mildew are highly coveted by rose producers. Until now, a limited number of studies have focused on finding a practical use of light quality to control powdery mildew on Rosa, which is one of the most frequent and serious diseases of greenhouse-grown roses. The manuscript has been submitted to the “Plants” journal, special issue “Applications and Advances in Artificial Light for Horticulture and Crop Production" because of the practical possibilities of using our results in horticulture.
The reviewer suggests that the paper should be corrected by a native speaker. I would like to point out that I am not a native speaker and therefore this article was sent for a language proofreading. The manuscript has been professionally proofread by Proof-Reding-Service.com. LTD, United Kingdom (the certificate is attached).
The results and discussion part has been improved. Irrelevant and outdated literature references have been removed and this chapter has been supplemented with the latest achievements as suggested by the reviewer. The obtained results were justified in more detail, especially in the chapter "Chlorophyll, flavanols and nitrogen balance indices".
Keywords have been supplemented with more precise words.
In introduction chapter, the latest statistical worldwide data concerning flowers and potting roses was provided. Moreover, irrelevant literatures was removed. Now, this chapter better suits the title of the manuscript ant to the aim of the study. In our opinion, future prospects of the study for flower producers are provided in the last sentence of “Introduction”.
Reviewer 2 Report
The author's research has application value. But I think it needs improvement. My suggestion is as follows:
Results and Discussion
p140-147 Flower buds are produced after one week of light treatment? The plant condition should be described in Materials and Methods.
p153-163 The description is not clear enough. The results of the study are generally consistent with those of other studies, but in the previous studies "increased R: FR reduced the number of flowers". Increase the number of flowers after 6 weeks in your study.
Fig 6. Why are there no data of the non-destructive index in the 3 and 4 weeks?
Fig 5 and 6. CHL did not increase but Fv/Fm increased significantly in S1 and S2 treatments. The author should discuss it.
Materials and Methods
p288-292 The light treatment is not written clearly enough ex. “3) W-white (cool-white diodes, 5,000 K) and two different 290 colour-mixing R (as dominant colour), B, W and far-red (730 nm ), such as 5) S1 – RBW".
P301-304 Table1. Why do five LED light treatments have no or little UV? Isn't it a supplemental light treatment?
Disease pathogens exist in the greenhouse. Are all the light treatments a disease condition in this study? The mixed analysis of light and heavy infections will make us unable to understand the impact of real light treatments. The author must explain the treatment in more detail.
Author Response
First of all, I would like to thank the referee whose suggestions have definitely improved the paper.
As suggested by the reviewer, plants condition has been described more precisely.
Incorrect description of the results related to the R: FR ratio has been corrected.
Unfortunately, we did not perform the measurements of chlorophyll content index and FV/FM after 1 and 3 weeks of plant growth, but we hope that this does not significantly affect the interpretation of the results.
The descriptions of the light treatments have been improved and detailed information on the individual spectra has been provided in Table 1 and Fig. 8. The light spectra presented in the graph and table concerned only the additional LED illumination and not their combined used with sunlight. Therefore, they contained little UV.
In our study, roses at the same stage of development were transferred to the greenhouse chamber with high Podosphera pannosa inoculum potential. The experiment was carried out in a greenhouse chamber where ‘Aga’ roses were grown previously. Moreover horizontal airflow fans were used to ensure air mixing and a stable spore distribution over the course of the experiment. So, the conditions for powdery mildew development were the same for all light treatments. We agree with the reviewer's opinion that the mixt analysis of light and heavy infections makes it difficult to understand the real contribution of light. However, in our opinion, this would require the use of a greater number of varieties with different resistance to powdery mildew. Such studies could be carried out in the future.
Reviewer 3 Report
In general, the manuscript is well written and presented. It is very well organized.
Some comments are provided regarding some suggestions about how the manuscript might be improved.

Author Response
First of all, I would like to thank the referee whose suggestions have definitely improved the paper.
All the reviewer's suggestions were presented in the manuscript in the “review mode”. These were mainly technical errors and obvious our mistakes.
Chapter “Material and Methods" has been supplemented with information how the plants were isolated among the light treatments (white lightproof screens).
Figure 8 has been improved by merging 6 small graphs into one graph and adding treatment names for better readability. Unfortunately, the graphs are generated by the application of the spectrometer and it is not possible to change the resolution. Charts could be clearer if they were larger. But it depends on the Journal's editors. Unit names (chlorophyll content index and flawonol index) have been improved.
All correction in manuscript have been added in the “review mode”.
Round 2
Reviewer 2 Report
Accept in present form